# PREDICTION OF TOURISM FLOW WITH SPARSE DATA INCORPORATING TOURIST GEOLOCATIONS

## ABSTRACT

Modern tourism in the 21st century is facing numerous challenges. One of these challenges is the rapidly growing number of tourists in space-limited regions such as historical city centers, museums, or geographical bottlenecks like narrow valleys. In this context, a proper and accurate prediction of tourism volume and tourism flow within a certain area is important and critical for visitor management tasks such as sustainable treatment of the environment and prevention of overcrowding. Static flow control methods like conventional low-level controllers or limiting access to overcrowded venues could not solve the problem yet. In this paper, we empirically evaluate the performance of state-of-the-art deep-learning methods such as RNNs, GNNs, and Transformers as well as the classic statistical ARIMA method. Granular limited data supplied by a tourism region is extended by exogenous data such as geolocation trajectories of individual tourists, weather and holidays. In the field of visitor flow prediction with sparse data, we are thereby capable of increasing the accuracy of our predictions, incorporating modern input feature handling as well as mapping geolocation data on top of discrete POI data.

## 1 INTRODUCTION

With increasing population and travel capacities (e.g. easy access to international flights) cultural tourism destinations have seen a rise in visitor counts. In addition, recent needs for social distancing and attendance limitations due to the global COVID-19 pandemic have confronted tourism destinations with significant challenges in e.g. creating and establishing sustainable treatment of the both urbanised and natural environment or e.g. preventing overcrowded waiting-lines. The perceptions of tourists regarding health hazards, safety and unpleasant tourism experiences may be influenced by social distance and better physical separation Sigala (2020). As far as The United Nation's 2030 Agenda for Sustainable Development UNWTO (2015) is concerned, tourism has not only the potential to contribute to several of the 17 Sustainable Development Goals (SDGs), but moreover an obligation. Only by establishing sustainable tourism it will be possible to create

- sustainable cities and communities (Goal 11)
- responsible consumption and production (Goal 12)
- decent work and economic growth (Goal 8)

Therefore, future-oriented tourism regions aim to first *understand* and then *control* visitor flows in order to

- preserve and protect their natural landmarks
- reduce emissions and waste as a result of overcrowding e.g. in parks or narrow city centers
- establish sustainable energy consumption within tourist attractions
- create harmony between residents and tourists
- and maximise tourist satisfaction, which is directly connected to the economical wealth of the specific tourism region.

Unfortunately, many real-world problems suffer from sparse data availability due to data compliance issues, lack of data collection or even lack of data transfer through stakeholders. In the end there are

not enough datasets to properly train state-of-the art machine learning models. On the other hand there are datasets available, where ethical considerations have to be made on whether they full-fill data privacy policies as well as comply the rights of tourists. In our research we use non personal data collected by POIs, tourist or tourist related facilities as well as anonymized digital device data. Although tourists agree on e.g. sharing entry timestamps or locations, the collection of this data is often a side product of services such as ticket sales, travelcards or digital apps. The latter is the most controversial dataset used in this research, since it is location data collected by Mobilephone-Apps. The dataset can be compared to Mobilephone-data collected from mobile network operators, which displays locations of devices throughout a specific time-period. This common practice collection of data is entirely profit-oriented, since the companies collecting this data specifically aim to sell it. Apart from the fact that such data can help to improve scientific research and overcome real-life problems, it has to be discussed whether people are aware of what they are sharing by using these services, even if these datasets do not contain direct personal data.

The question on how to improve awareness of data shared by such apps or services is not answered in this research. This scientific work is focusing on what is possible to achieve in the given environment considering the given data and data history in regards to tourist flow prediction, since sparse data is a wide spread generic problem.

The first step in order to control tourist flows is to predict authentic movement and behavior patterns. However, since the tourist visitor flow is affected by many factors such as the weather, cultural events, holidays, and regional traffic and hotspots throughout a specific day, it is a very challenging task to accurately predict the future flow Liu et al. (2018). Due to the availability of large datasets and computational resources, deep neural networks became the state-of-the-art methods in the task of forecasting time-series data Pan et al. (2021), including tourism flow applications Prilistya et al. (2020).

In this work, we focus on tourist flow prediction based on a local dataset from the visitors of the tourist attractions of the city of Salzburg as well as third-party geolocation data of individual tourists. After data preprocessing and dataset preparation, we attempt to compare the performance of different deep-learning based methods for time-series prediction with ARIMA, a traditional statistics based method. According to Li and Cao Li & Cao (2018), ARIMA is the most popular classical time forecasting method based on exponential smoothing and it was made popular in 1970s when it was proposed by Ahmed and Cook Ahmed & Cook (1979) to be used for short-term freeway traffic predictions.

Deep neural networks are proven to work very well on large datasets. However, their performance can degrade when trained on limited data, resulting in poor predictions on the test set. Since limited data is a common problem in tourism time-series forecasting, we perform a comprehensive comparison of the DNNs and traditional techniques on a small dataset to reveal the shortcomings and point out necessary future improvements.

**DL based models.** Recurrent Neural Networks (RNNs) are the state-of-the-art models for learning time-series datasets. RNNs equip the neural networks with memory, making them successful at predicting the sequence-based data. The introduction of gating mechanism to RNNs lead to the great performance of LSTM Hochreiter & Schmidhuber (1997) and GRU Chung et al. (2014).

RNNs have limitations when facing irregularly-sampled time-series, present at many real-world forecasting problems such as tourist flow prediction. To address this limitation, phased-LSTM Neil et al. (2016) adds a time gate to the LSTM cells. GRU-D Che et al. (2018) incorporates time intervals by a trainable decaying mechanism in order to handle time-series with missing data and long-term dependencies.

Another approach is to introduce time-continuous models with latent state defined at all times such as CT-RNN ichi Funahashi & Nakamura (1993), CT-LSTM Mei & Eisner (2017) and CT-GRU Mozer et al. (2017). A family of continuous-time networks are NeuralODEs Chen et al. (2018) that define the hidden state of the network as a solution to an ordinary differential equation. Some limitations of NeuralODEs such as non-intersecting trajectories can be aleviated by using augmentations strategies leading to Augmented-NeuralODEs (ANODEs) Dupont et al. (2019). Continuous-time models share some favourable properties: Adaptive computation as they can be implemented by numerical ODE (ordinary differential equations) solvers and training with constant memory cost by using the adjoint sensitivity method Chen et al. (2018). In addition, they can be statistically verified

by using GoTube Gruenbacher et al. (2022) which constructs stochastic reachtubes (=the set of all reachable system states) of continuous-time systems and is made deliberately for the verification of countinuous-depth neural networks.

Transformer-based models Vaswani et al. (2017) have achieved great success in many tasks, starting from natural language processing Devlin et al. (2018) to computer vision applications Dosovitskiy et al. (2020). Due to their great capability on sequence learning and representation, many recent research have explored their performance on time-series forecasting tasks, especially for datasets with long sequences and high historical information Zhou et al. (2020); Wu et al. (2021); Zhou et al. (2022). Multi-head self-attention mechanism is the main component of transformer models, enabling them to extract correlations in long sequences. However, self-attention is permutation-invariant, and can lead to loss of temporal dependencies.

Graph Neural Networks (GNNs) are an interesting new class of Deep Learning Algorithms that allow for the inputs to be structured as graphs. Most GNN models build on the notion of Graph Convolutions which can be seen as a generalization of Convolutional Neural Networks to graph structured data - as opposed to being arranged in a grid. An even more fascinating type of DL model are temporal GNNs that combine Graph Convolutions with RNNs. Such temporal GNN models are most prominent in traffic flow prediction applications. Zhu et al.Yu et al.Guo et al.Bai et al.Li et al.

**Traditional techniques.** For time-series forecasting with traditional techniques we use the Autoregressive Integrated Moving Average (ARIMA) model. ARIMA has been used in recent studies as a baseline for the evaluation of novel deep-learning based models Yao & Cao (2020); Bi et al. (2020); Li & Cao (2018); Hassani et al. (2017) and is thus selected as a baseline model for this paper as well.

Since limited data is a common problem in tourism time-series forecasting, we summarize the specific contributions of our paper as follows:

- We perform a comprehensive comparison of the DNNs and ARIMA, a traditional technique, on a real-world dataset to reveal the shortcomings and point out necessary future improvements.
- The real-world dataset is considered small because of limited historical entries.
- Per point-of-interest (POI), we perform granular predictions on an hourly basis, which is critical for the task of tourism flow control.
- We further evaluate modern DL techniques such as Transformers and GNNs.
- To the best of our knowledge, we are the first to apply a wide range of DL models to tourist flow prediction.

## 2 RELATED WORK

Due to the importance of tourist flow prediction in the ever-growing tourism industry, visitor forecasting has gained some attention over the past years Burger et al. (2001); Hassani et al. (2017). Existing work examines tourist demand forecasting using Recurrent Neural Networks such as LSTMs Rizal & Hartati (2016); Li & Cao (2018) or hidden Markov Models in conjunction with deep neural networks  Yao & Cao (2020). Most of these studies make predictions with only a limited set of models.

Data granularity is another important aspect of tourism data. Many studies focus on long-term predictions of monthly, quarterly and yearly, or in the best case daily number of visitors Bi et al. (2020) in large regions as city or country-level tourism demand Asvikarani et al. (2020). However, performing granular predictions on an hourly basis and per POI is critical for the task of tourism flow control.

### 2.1 PREDICTION OF TOURISM FLOW

Due to the importance of tourist flow prediction in the ever growing tourism industry, visitor forecasting has gained some attention over the past years Burger et al. (2001); Hassani et al. (2017). Existing work examines tourist demand forecasting using Recurrent Neural Networks such as LSTMs

Rizal & Hartati (2016); Li & Cao (2018) or hidden Markov Models in conjunction with deep neural networks Yao & Cao (2020). Most of these studies make predictions with a limited set of models.

Data granularity is another important aspect of tourism data. Many studies focus on long-term predictions of monthly, quarterly and yearly, or in the best case daily number of visitors Bi et al. (2020) in large regions as city or country-level tourism demand Asvikarani et al. (2020). However, performing granular predictions on an hourly basis and per POI is critical for the task of tourism flow control.

## 3 DATA

Two different data sources were combined to enable the use of their different features in the training of the models and prediction of future visitor counts.

The first dataset we used stems from the "Salzburg Card" which was kindly provided to us by TSG Tourismus Salzburg GmbH. Upon purchase of these cards, the owner has the ability to enter 32 different tourist attractions and museums included in the portfolio of the Salzburg Card. The dataset consists of the time-stamps of entries to each POI. Additionally, we used data about weather and holidays in Austria. The *Salzburg Card dataset* will be published alongside this paper in normalized and hourly aggregated fashion. A more detailed description of the dataset can be found in the Appendix.

In addition we used location data gathered through mobile phone applications of visitors to the city of Salzburg by a third-party service. This dataset roughly covers app. 3% of tourists and helps us in better evaluating tourists inbetween POI's. This dataset relies on location data transmitted by mobile-phone apps and contains sparse data about tourist-locations without any destinct frequency of recording. Since we are predicting tourism flow in the city of Salzburg it comes naturally to incorporate the street graph which can be obtained from OpenStreetMap (OSM) OpenStreetMap contributors (2017). By constructing a bounding-box around the city center containing the majority of POIs we can then query the OSM graph using the `osmnx` python package. The resulting graph contains 2064 nodes for the crossings and 5359 edges for the streets where edge values correspond to the lengths of the street segments. In order to incorporate the location tracking data we mapped it to the OSM graph. Each reported location was mapped to the nearest node of the graph followed by aggregating the total number per hour.

**CoVID-19** Tourism around the globe saw huge drops during the global CoVID-19 pandemic. Starting in march of 2020 Austria started to take preemptive measure to prevent the spread of the virus. These travel restrictions and closings of public spaces, hotels and restaurants severely reduced the number of tourists in and around the city of Salzburg. As a consequence, prediction accuracy could be diminished when using models that have been trained on pre-CoVID data.

## 4 METHODS

For this work, we built our own dataset on hourly data collected from tourist attractions and then expanded this by including geolocation data. Including many different datasources is a key challenge for this real-world prediction task. Sparse geolocation data is therefore fed into our GNN model as features. With this approach we aim to create models that are capable of easily integrating new datasources that might be available in the future. We then perform predictions with a rich set of models and do a comprehensive comparison of the results. In this section we first introduce the dataset we used for the experiments. Then we go over the methods we chose to evaluate and compare their performances.

### 4.1 DEEP-LEARNING MODELS

We use a large set of RNN variations on the tourist-flow dataset to perform a comprehensive comparison of the state-of-the art models and provide insight on their performance. The set comprises vanilla-RNN, LSTM, phased-LSTM, GRU-D, CT-RNN, CT-LSTM and Neural-ODE networks. Moreover, we used a Transformer model, using only the encoder part with 8 heads, 64 hidden

units, and 3 layers, to forecast the tourist flow. Finally, we applied Attention Temporal Graph Convolutional Networks (A3T-GCN) Zhu et al. to our prediction problem in order to utilize geolocation data of individual tourists.

All of the Neural Networks were trained with Backpropagation-Through-Time and the Adam optimizer Kingma & Ba (2014) using the parameters given in the Appendix 3. In order to find optimal model size, loss function, and whether to use normalized visitor counts, we did a grid search conducting three runs per configuration and keeping the one which achieved the lowest RMSE. The loss functions listed in Table 3 correspond to Mean-Squared-Error, Mean-Absolute-Error and Huber-Loss respectively.

## 4.2 TRADITIONAL METHODS

In this study we use the non-seasonal variant of the ARIMA model which does not consider the seasonal patterns in a time-series. This model is usually denoted as *ARIMA (p,d,q)*, where $p$ is the number of autoregressive terms, $d$ is the number of non-seasonal differences, and $q$ is the number of lagged forecast errors in the prediction equation Burger et al. (2001). A generic expression for the non-seasonal ARIMA process is given as Hyndman & Khandakar (2008):

$$\phi(B)(1 - B)^d y_t = c + \theta(B)\varepsilon_t$$

where $\{\varepsilon_t\}$ is a white noise process, $B$ is the backshift operator, and $\phi(z)$ and $\theta(z)$ are polynomials of order p and q respectively. The parameters $p$, $d$, and $q$ define how the function will fit the given time-series which directly affects the quality of future predictions. There are several recommended approaches for the manual selection of parameters and they all rely on comparing the ARIMA model fit to the real values of the time-series to try and minimize the deviation between them Ahmed & Cook (1979); Hyndman & Khandakar (2008); Hyndman & Athanasopoulos (2018).

For this study, we have used the *auto.arima* function from the *R forecast* library that automatically fits the ARIMA model with different sets of parameters and returns the best combination of $p$, $d$, and $q$ for the given time-series Hyndman & Khandakar (2008); Hyndman & Athanasopoulos (2018). Given that each of the 32 POIs in this study has a different time-series of visitor counts, we have determined different ARIMA parameters for each POI which were the best fit for its time-series. ARIMA predictions models were then built individually for each POI by fitting the ARIMA to that POI's training dataset with the best $p$, $d$, and $q$ parameters for that POI using the Python *pmdarima* library. Each time after the number of visitors is predicted for the next hour in the test data, the true value (i.e., number of visitors) for that hour is added to update the existing ARIMA model and make it aware of all previous true values before making the next prediction.

## 4.3 PREPROCESSING

We used the Salzburg card data from years 2017, 2018, and 2019 for our first set of experiments. In order to create the time-series data, we accumulated the hourly entries to each location. The data then consists of the hour of the day, and the number of entries at that hour to each of the 32 POIs.

For the DL models, we added additional features to the dataset: Year, Month, Day of month, Day of week, Holidays and Weather data. For the Holiday data we used the national holidays and school holidays and count the days to the next school day. For the Weather data, we use the hourly weather data with these features: Temperature, Feels Like, Wind speed, Precipitation, and Clouds as well as a One-Hot-Encoded single word description of the weather (e.g. "Snow").

We performed further pre-processing by normalizing all features to values between $0$ and $1$. To account for seasons, we performed sine-cosine transformation for the month. Intuitively, since it is a circular feature we do not want to have the values for December and January to be far apart.

Finally, We split the data into sequences of length 30, and used the data from years 2017 and 2018 as the training set, and 2019 as the test set.

**Graph Neural Networks** For the GNN we used the OSM graphs as illustrated in Section 3. Our dataset of tourist locations was very sparse which subsequently resulted in very sparse inputs for each node. Since we are trying to predict numbers of entries at the POIs, we added them as additional nodes to the graph connecting them to up to 5 of the nearest nodes present in the graph with a max

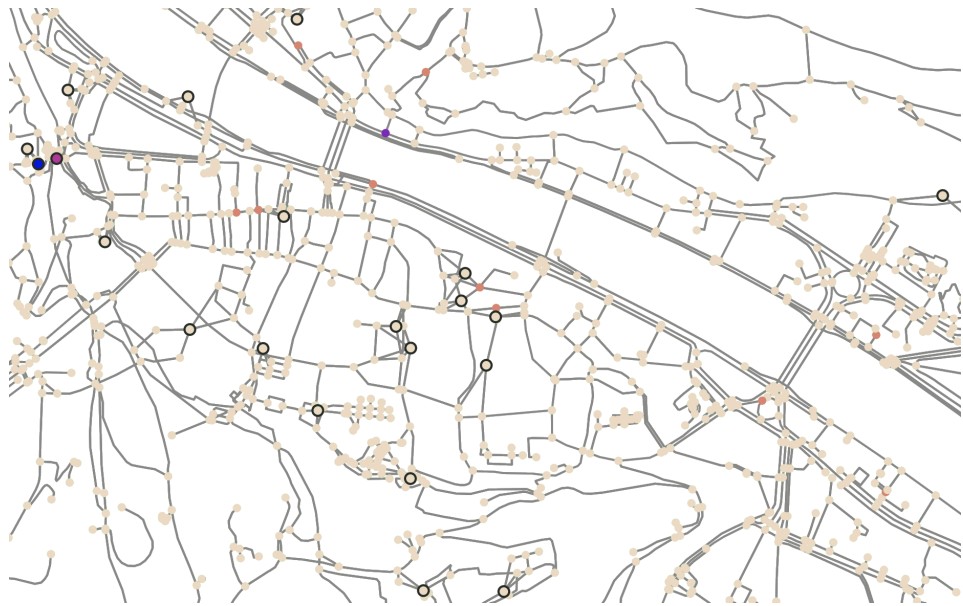

Figure 1: One sample of the series of OSM graphs of the Salzburg city center obtained from preprocessing. Encircled nodes are the special POI nodes. Color coded are the normalized aggregated entry and tracking data, where most of the nodes indicate zero (pale)

distance of 80 m. Finally, the global features such as weather and holidays are added to the graph by a linear mapping from features to nodes. Each node received a second feature that represents the influence of these global features. This way we obtained a series of graphs where each sample constitutes the OSM graph with the edge values corresponding to the distance and the node values corresponding to the aggregated location data and POI entries as well as a linear combination of the global features. One sample is visualized in Figure 1.

For training we used both the location and POI data-sources as labels by computing a weighted sum giving more importance to the POI entry data $loss = 0.9\ l_{poi} + 0.1\ l_{other}$. For $l_{poi}$ and $l_{other}$ we chose For inference we predicted the whole graph and discarded the nodes that do not represent POIs. Our results show partial success for the GNN approach since it was able to handle the very sparse tracking data which led to overfitting in all other models.

| Model | # Cells / # Parameters | Time Train (min) | Pred (ms) | only visitors MAE | RMSE | external features MAE | RMSE |
|---|---|---|---|---|---|---|---|
| ARIMA | **224** | - | 69k | 5.217 | 7.833 | - | - |
| ANODE | 64 / 21.3k | 145.6 | 3.01 | 4.599 | 6.965 | 4.410 | 6.663 |
| Vanilla RNN | 128 / 43.7k | 5.9 | **0.18** | 3.958 | 6.321 | 3.802 | 6.160 |
| LSTM | 32 / 11.9k | **1.5** | 0.24 | 3.713 | 6.209 | 3.630 | 6.113 |
| Phased LSTM | 32 / **11.8k** | 27.0 | 0.46 | 3.825 | 6.359 | 3.651 | 6.120 |
| CT-LSTM | 32 / 19.9k | 18.1 | 0.31 | 3.734 | 6.239 | 3.700 | 6.185 |
| CT-RNN | 128 / 27.4k | 57.1 | 0.60 | 3.694 | 6.131 | 3.629 | **5.983** |
| GRU-D | 64 / 27.7k | 16.6 | 0.33 | **3.638** | **6.121** | **3.621** | 6.073 |

Table 1: The prediction results for models when using only the visitors count data and when using additional features from external data. Our experimental results show that the errors were smaller for RNNs compared to ARIMA both with and without additional features were used.

## 5 MAIN RESULTS

### 5.1 FIRST SET OF EXPERIMENTS

We performed a diverse set of experiments with ARIMA and DL models to evaluate their forecasting accuracy, execution time and prediction time and compare the models. Table 1 shows the Mean-Absolute-Error (MAE) and Root-Mean-Squared-Error (RMSE) achieved for each method applied to the timeframe from 2017-2019 - before COVID.

For the deep-learning models model size, corresponding number of parameters and training and prediction times of the best run are listed additionally. Since using normalized visitor counts led to better results in every single model, we omitted non-normalized models from the table. All of the models except the ANODE achieved best results when trained with MAE as the loss function. For ANODE huber loss was best. The phased LSTM uses the least number of parameters for comparable results.

All of our deep-learning models were able to outperform the ARIMA method in both metrics, with and without additional features. Providing the additional features to the models resulted in a slightly better performance for the DL models. The improvements were quite marginal because we have limited training data available and thus increasing the number of features does not pay off, or even might result in over-fitting. ARIMA was not able to use the external features and fails when given only short sequences as input. The DL models can handle shorter sequences when trained on the whole training set. Hence, in order to ensure the fairness in our comparisons, we report the results with and without using the additional features of weather and holidays for DL models.

We measured training times for the DL models and prediction times for all models on a standard workstation machine setup (4 CPU cores with 3.9 GHz, 8GB DDR3 memory) equipped with a single GPU (GeForce GTX 1050 Ti, 4GB memory). As shown in Table 1 ARIMA took 69s to perform a single prediction for all POIs and the DL took for the same task fractions of milliseconds, while having used once y minutes for training. It is not possible to directly compare training and testing runtimes between them because ARIMA does not have a dedicated training step where the model is built. Instead, before every new prediction at time t, ARIMA model is fitted to the known time series up until the time t-1 (i.e., training and testing are integrated for every prediction). Therefore, our integration of ARIMA favors accuracy over prediction time. ARIMA calculations are timeconsuming, since it makes predictions for each POI separately, while the DL models are trained with the visitors to all POIs in a single vector and make predictions for all at the same time. Thus, ARIMA loses the implicit data about the total number of visitors in the city.

In order to visually explore the predictions made by the models, we plotted the predictions and the ground truth for a few selected time-windows (see Figure 2). We plot the predictions made by the DL models (including the external features) with the best MAE and RMSE, which were the GRU-D and CT-RNN respectively. The prediction made by the DL models with the visitors only data was only slightly worse than the others, which is why we omit these evaluations in the plots. Our plots show that although ARIMA is out-performed by the DL methods in the average error of all predictions, there are cases where it actually performs better than the other models. The plot on the Top shows the forecast and real values for the tourists entered the Funicular Railway descend which is the cable car ride leading up to Salzburg Castle. As visible in the plot, the DL models show a better performance, especially in the valleys where the ARIMA fails to predict the downfalls accurately. Mid shows visitor predictions for Mozart's Birthplace Museum around the time of New Year's Eve. The reduced numbers of visitors on the 1st and 2nd of January is overestimated by all our models. Finally, on the bottom the predictions for the Festival Hall Festspielhaus guided tour are shown which is sparse since it takes place once a day at 2 pm. All models fail in prediction for the second and third peak at this location. However, CT-RNN shows a very good performance in predicting the first and last peak and at least shows an upward trend for the second and third peak. ARIMA can not handle this type of sparse data at all.

### 5.2 EXTENDED SET INLCUDING GEOLOCATION DATA

We conducted a second set of experiments on the timeframe from 2019 to 2021 since only for this period geolocation data of individual tourists was available. Results are presented in Table 2 which shows for each model the number of Parameters and MAE with and without additional features and

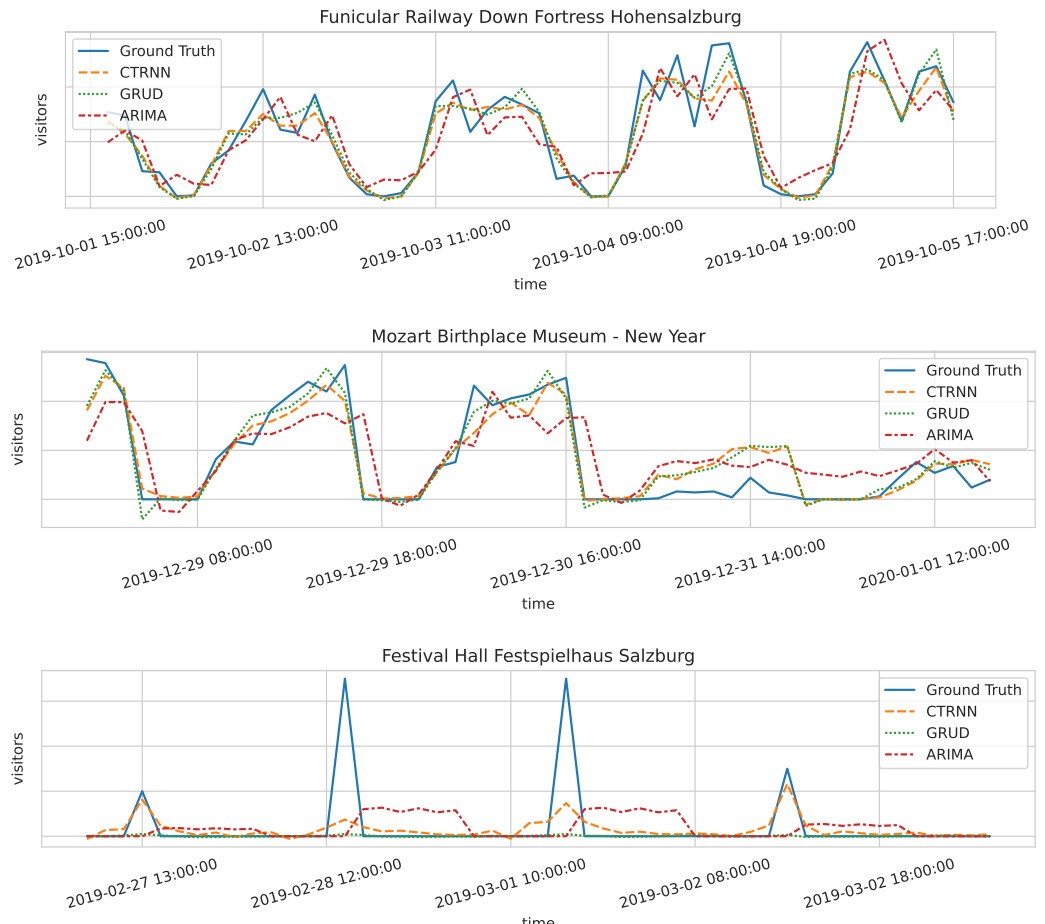

Figure 2: Predicted and True visitor counts for the Funicular Railway (top) and Mozart's Birthplace Museum (mid) and the Festival Hall (bottom). Predictions are computed using CT-RNN (orange), GRU-D (green) and ARIMA (red).

also when using features and the geolocation data. This time we included the Transformer and GNN models, but excluded ARIMA and ANODE due to computation time reasons. Since the *Salzburg Card dataset* for this particular timeframe contains a significantly lower number of datapoints due to lockdowns enforced by the government, the numbers must not be compared directly to the results discussed in the last section.

Relatively speaking, the RNN models show comparable prediction accuracy although Transformers led to the best results. Transformers can handle multi-variate data well due to the multi-head self-attention mechanism which enables them to extract hidden correlations in input, and hence get better loss after using additional features. However, they require considerably more parameters in comparison to the RNN models.

For GNNs we only conducted experiments with additional geolocation data since input graph attributes would be even sparser defeating the point of using a graph based approach. The A3T-GCN algorithm scored a slightly worse prediction error in comparison. However, all other methods failed to converge when trained on the sparse geolocation data which shows the usefulness of the GNN approach.

**Only with GNNs we are able to utilize sparse geolocation Data into meaningful predictions.** Since there is more sparse geolocation data expected to be processed within real-life-scenarios, this is the only approach to fit these needs. Also we have to note that no explicit Hyperparameter tuning was conducted which means that the GNN prediction accuracy could be further approved upon.

| Model | # Parameters | MAE | | |
|-------|-------------|-----|-----|-----|
| | | only visitors | + features | + additional geolocation |
| Vanilla RNN | 11.0k | 2.75 | 2.58 | 430.88 - diverging |
| LSTM | 23.6k | 2.49 | 2.41 | 422.63 - diverging |
| Phased LSTM | 23.4k | 2.56 | 2.40 | 426.89 - diverging |
| CT-LSTM | 39.9k | 2.51 | 2.40 | 428.47 - diverging |
| CT-RNN | **7.0k** | 2.54 | 2.48 | 431.74 - diverging |
| GRU-D | 27.7k | 2.53 | 2.46 | 431.71 - diverging |
| Transformer | 253k | **2.35** | **2.23** | 482.78 - diverging |
| A3T-GCN | 39.7k | - | - | **2.71** |

Table 2: The prediction results for 2021 after training on data from 2019 & 2020. Each model (except the GNN) was trained on just the visitors, visitors and external features and visitors with both features and tracking data. Reported is the best MAE on the validation set for three different initializations. Using the sparse tracking data led to divergence in all models except GNNs.

# 6 CONCLUSIONS AND FUTURE WORK

We performed a thorough evaluation of deep-learning (DL) versus ARIMA, a traditional method, on the task of forecasting tourist flow time-series. We found that all of the DL models were able to outperform the established ARIMA method when using only visitor counts as the training data. Extending the dataset with additional features of time, holidays, and weather improved the predictions of the DL models, while ARIMA is not able to handle additional features. While the improvements were little, the performance might still get boosted by performing feature selection for the weather data and improving the holidays features. In terms of the prediction time, DL models are meaningfully faster than ARIMA, which is an important aspect since most real-world applications require fast inference time.

We then evaluated various DL techniques including Transformer and GNN on the timeframe where COVID hit while also incorporating sparse tracking data from mobile phones. The results showed that adding external features again led to better results. Hoewever, when adding the geolocation data, all of the RNN models including the Transformer diverged. The A3T-GCN model on the other hand, was able to deal with this additional data albeit resulting in a slightly higher validation error. This work therefore serves as a proof-of-work for incorporating spatial structure using GNNs that is robust to overfitting even when fed sparse data and prediction accuracy should improve a lot when denser geolocation data is available.

This paper opens many avenues for future research. The most straightforward next step is to try to improve the performance of the DL methods using additional training techniques such as regularization or learning rate scheduling. Admittedly, the ARIMA method is most suitable for predicting univariate data which is why our classical methods should be extended by also considering methods for multivariate forecasting such as Vector autoregression (VAR). Furthermore, there also exist online methods (e.g. Online ARIMA Liu et al.) that allow for adjusting the model for new datapoints.

In addition, we want to use the knowledge gained to build specialised models which outperform state-of-the art models in terms of short-term prediction with limited data. Another direction is to work on predictions for a longer time horizon and incorporate these predictions into a recommender system for tourists. Therefore, we would like to go one step further into the direction of giving tourism regions the ability to control the visitor flow.

# 7 REPRODUCIBILITY STATEMENT

In order to ensure reproducibility we publish the *Salzburg Card dataset* alongside this paper. The tracking data used for the second set of experiments can not be published due to copyright. We will also hand in our sourcecode with the paper and make it available on Github.

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

# A  APPENDIX

## A.1  DATA

The dataset we used stems from the "Salzburg Card" which was kindly provided to us by TSG Tourismus Salzburg GmbH. Upon purchase of these cards, the owner has the ability to enter 32 different tourist attractions and museums included in the portfolio of the Salzburg Card. We accumulated the hourly amount of entries into these points of interest(POI's).

Every entry from 2017 to 2021 using thses cards was recorded and summed up in this dataset. 2017 to 2019 serves as a baseline of "normal" tourism activity in the city of Salzburg. The data from 2020 and 2021 is of particular interest since the visitor numbers were heavily influenced by the global CoVID-19 pandemic. This affects overall visitor numbers as well as distributions among the different attractions due to travel and local restrictions.

**Name.** Name of the POI
**Timestamp.** ISO 8601 timestamp. Accumulated to visitor entries per hour, if no visitors were recorded for a certain timeframe the entry was omitted.
**Visitors.** Amount of card users entering the venue in the next hour. Not all visitors need to use this card so this data represents a subset of all tourists visiting these POI's.
**Longitude.** Lonitude of the visited POI
**Latitude.** Latitude of the visited POI

## A.2 HYPERPARAMETERS

| Hyperparameter | Value |
|---|---|
| sequence length | 30 |
| batch size | 16 |
| epochs | 300 |
| optimizer | adam |
| Learning-rate | $1e^{-3}$ |
| $\beta_{1,2}$ | $(0.9, 0.999)$ |
| $\epsilon$ | $1e^{-8}$ |
| loss function | mse, mae, huber |
| model size | 32, 64, 128 |
| normalized visitors | True, False |

Table 3: Hyperparameters used in RNN training.

