# OpenReview forum: "PREDICTION OF TOURISM FLOW WITH SPARSE DATA INCORPORATING TOURIST GEOLOCATIONS"
_ICLR.cc/2023/Conference — Submitted to ICLR 2023_

### Official Review · Reviewer_XQLY · 2022-10-18

**Confidence:** 3
**Correctness:** 2
**Technical Novelty And Significance:** 2
**Empirical Novelty And Significance:** 3
**Recommendation:** 3

**Clarity, Quality, Novelty And Reproducibility:**

* Clarity & Quality: The paper has typos, duplicate text, and other formatting errors. See the "typos" section below. The figures and tables are generally in good shape.
* Novelty: The paper does not propose a novel method, but does provide a new dataset and new experimental results on that dataset. I have no objections to the paper on the grounds of novelty.
* Reproducibility: The paper trains 7 deep learning models but only provides a handful of hyperparameters in the appendix - I suspect more information is needed for reproducibility. The paper promises to release code and data.

Minor comments:
* Consider moving the discussion of RNNs / LSTMs / GNNs / etc. from the introduction to the related work sections.
* "The backshift operator" is not defined on page 5.
* The "POI" acronym is not defined before it is used in the abstract.
* It would be nice to see an ablation study showing the relative importance of the different "external features" (sparse location data, weather, holidays).
* Please be careful to separate speculation from real claims, e.g. "The improvements were quite marginal because we have limited training data available and thus increasing the number of features does not pay off, or even might result in over-fitting." is stated definitively, but seems to be speculation.

Typos:
* "full-fill" should be "fulfill" on page 2
* "Mobilephone-Apps" should be "mobile phone apps" on page 2.
* "Mobilephone-data" should be "mobile phone data" on page 2.
* There is duplicate text at the bottom of page 3 / top of page 4.
* "march" should be "March" on page 4.
* "In this section we first introduce the dataset..." on page 4 does not accurately reflect the order of what is to follow.
* "destinct" should be "distinct" on page 4.
* There are more, but I stopped keeping track.


**Details Of Ethics Concerns:**

Given the use of individual tracking data (even though that data will not be released) and the goal of modeling and predicting human activities, it seems reasonable to think about privacy concerns, related legal issues, and applications in mass surveillance.

**Strength And Weaknesses:**

# Strengths
* The paper addresses an interesting spatiotemporal prediction problem.
* The paper attempts the important work of benchmarking a diverse collection of methods on a difficult task.
* The dataset which this paper will publicly release will no doubt be useful to the community, especially since it has data for before and during COVID.
* Investigating the use of side information (sparse geo-location information from tourists and metadata about weather and holidays) is worthwhile and interesting.


# Weaknesses
* The paper trains 7 different deep learning approaches with the same learning rate and reports the results. I believe it is uncontroversial to claim that tuning the learning rate is perhaps the most important step in training a deep learning algorithm. Given this, it is difficult to trust the results in the paper.
* It is not clear whether a validation set was used or (if a validation set was used) how it was constructed. Also, which split is used for model selection?
* The ARIMA model gets to use the true values for past time points before making a prediction. Do the deep learning models get a similar advantage? If so, how is it implemented?
* The results are not contextualized. At what point is an improvement meaningful? Are MAE and RMSE effectively capturing what is important for the end-user? For instance, in Table 1 we see that GRU-D is better than CT-RNN by 0.01 RMSE when doing "only visitors" prediction. Is this a meaningful difference for the use case? Does this represent a stable difference between the models, or would the results reverse if we trained again?
* There is no discussion of the potential for bias in the data. Is there a good reason to believe that Salzburg Card holders are a representative sample of the population of tourists? How is the mobile phone data sampled from the population? If these datasets are biased, what effects could they have on the models?

**Summary Of The Paper:**

This paper considers the problem of predicting the number of tourists visiting different attractions on an hourly basis. Using data from 32 tourist attractions in Salzburg over a four-year time span, the paper compares 7 different deep learning approaches against a classical approach (ARIMA). The paper also explores the use of additional data (sparse geo-location information from tourists and metadata about weather and holidays).

**Summary Of The Review:**

This paper is tackling an interesting problem and producing an interesting dataset. With some changes, I think this paper could be great. My primary concerns relate to experimental methodology: on the basis of the paper, can we say anything confidently about which method works best? My claim is that currently the answer is "no" due to the problems listed under "weaknesses" above.

---

> ### Comment · Reviewer_XQLY · 2022-12-12
> **No response from authors**
>
> As there has been no response from the authors, my rating is unchanged.

---

### Official Review · Reviewer_xjA7 · 2022-10-23

**Confidence:** 4
**Correctness:** 3
**Technical Novelty And Significance:** 1
**Empirical Novelty And Significance:** 2
**Recommendation:** 3

**Clarity, Quality, Novelty And Reproducibility:**

This work is easy to follow but there are still parts that need further improvement using clear theoretical explanations. Moreover, the idea of the paper is just applying state-of-the-art deep learning methods compared with the classic statistical method to predict tourism flow and volume with limited novelty.

**Details Of Ethics Concerns:**

No ethics concerns.

**Strength And Weaknesses:**

Strength:

The prediction performance is compared on several deep learning methods on continuous sequential predictions.

Including external features to the times series dataset could slightly improve the DL models’ performance.



Weakness:

The authors mentioned “limited data is a common problem”; but no solution was discussed to overcome this challenge. Instead, Only supervised learning models were applied for training and prediction.

When comparing the model performance on time-series prediction problem, it would be great to further compare with the naïve prediction by simply setting all forecasts to be the value of the last observations, i.e., X_t = X_{t-1}.

In general, this work shows a comprehensive comparison across different SOTA models; however, the contribution on creating new models/approaches and theoretical insight is still missing.

Some contents in the introduction and related work look redundant.

**Summary Of The Paper:**

This paper evaluates the performance of state-of-the-art deep-learning methods such as RNNs, GNNs, and Transformers as well as the classic statistical ARIMA method to predict the tourism volume and tourism flow. Granular limited data supplied by a tourism region is extended by exogenous data such as geolocation trajectories of individual tourists, weather and holidays.

**Summary Of The Review:**

The article has introduced the idea of predicting the volume and flow of tourists by using discrete POI data, extra external factors, and geolocation data together with state-of-the-art deep learning techniques like GNNs, RNNs, and Transformers as well as traditional statistical ARIMA approaches.

---

### Official Review · Reviewer_mG9t · 2022-10-24

**Confidence:** 4
**Correctness:** 3
**Technical Novelty And Significance:** 1
**Empirical Novelty And Significance:** 2
**Recommendation:** 3

**Clarity, Quality, Novelty And Reproducibility:**

The paper needs some proof reading and some clarity before it is submitted again to a more appropriate conference. Terms are not defined, entire paragraphs are present twice in the paper, citations have problems, typos are numerous and could be removed with a spell-checker, etc.
The paper does not seem to present any novelty.
The data and the codes are or will be released.

Corrections:
abstract: define POI
full-fill -- fulfill
destinct - distinct?
CoVID --> Covid or COVID.
citation 'ichi Funahashi & Nakamura (1993)': problem in name?
Mobilephone-data not a word
repetition (p.3 and p.3/4): Due to the importance of tourist flow ... paragraph and  'Data granularity is another...' paragraph are shown 2 times, in 2. and 2.1.
transformer citation 'Zhu et al.': there is no date
4.2 equation: what is y(t)?
Figure 1: what are colors blue and red?
For lpoi and lother we chose For inference ... : sentence problem
Hoewever,


**Strength And Weaknesses:**

Strengths:
- the problem is interesting and worth investigating
- the data is released and could be useful for the community
- several methods were tested

Weaknesses:
- the paper does not propose a methodological novelty
- the problem is in the end very simple (single time series forecasting one step ahead), if we do not consider the geolocalization data with graphs (which currently does not work).
- use of MAE as the loss: this is unusual as MAE is not differentiable, so usually we use MSE.. why and how did you use MAE?
- the final goal is not clear, as the long introduction mixes Covid problems, increase of tourists problems (which are both unrelated), ethical problems due to goelocalizations bought to have more data.
- the paper does not seem to be novel with respect to state-of-the-art in this specific field, and experiments are not clear. Please state which are methods taken from previous works, and what is your method that would be novel. Is the statistical baseline, ARIMA, used in practice in this application? What is the gold standard today in tourist forecast?
- the forecast is only one step ahead, i.e. 1 hour, and it would be more interesting to augment this. Also, please indicate more clearly (i.e. from the start) what is the target (prediction of next hour entries).

**Summary Of The Paper:**

This paper presents a comparison of different deep learning methods for tourist flow 1-hour forecast in 30 touristic locations of Salzburg city. The data is an hourly entry number of visitors at each of the 30 sites during 3 years, with addition of some information on the day (holidays or not, etc.), and then a test using sparse tourist geo localizations is also presented at the end (but the results are worse). The deep learning methods for forecasting the time series (RNN, LSTM, GRU) are also compared with a 'standard' statistic method, called 'ARIMA'. The results are showing that deep learning models are indeed capable of better forecasting than ARIMA, but the addition of golocalization data was not useful.


**Summary Of The Review:**

While the application is of interest, the paper is not suited for publication at ICLR as no methodological novelty is presented, moreover, the application is rather simple and the paper needs important improvement and proof-reading.

---

### Official Review · Reviewer_X9rk · 2022-10-26

**Confidence:** 4
**Correctness:** 3
**Technical Novelty And Significance:** 1
**Empirical Novelty And Significance:** 2
**Recommendation:** 3

**Clarity, Quality, Novelty And Reproducibility:**

Clarity:

This paper is well-written with clarity. The authors’ description in this paper is clear.

Quality:

This paper is of low quality with little contribution and almost no novelty.

Novelty:

There is almost no novelty in this paper. The authors simply conduct several traditional methods and deep learning methods on the tourist flow prediction problem, while not proposing their own method.

Reproducibility:

This paper has high reproducibility, and the author uploaded the source code on Github.

**Strength And Weaknesses:**

Strengths:

1. The description of the problem is clear.

2. This paper is the first to apply deep learning methods to the tourist flow prediction problem.

Weaknesses:

1. The innovation of this paper is insufficient. Despite experiments that have been carried out on several methods, the authors do not propose their own methods.

2. The authors do not give a detailed explanation of why GNN methods seem to outperform other methods.

3. The experiment part in this paper is not enough. It lacks an ablation study part and a case study part.

**Summary Of The Paper:**

This paper uses discrete POI time series data to predict tourist flow. In this paper, the authors perform a comprehensive comparison between traditional methods such as ARIMA and deep learning methods such as RNNs and CNNs. As the first to apply deep learning methods to the tourist flow prediction problem, the authors point out the flaws and future research directions.


**Summary Of The Review:**

In summary, this paper should not be accepted. The authors simply carry out experiments that have been on the existing traditional methods and deep learning methods instead of giving their own solution to this problem. Besides, the author's analysis in the experimental part is also insufficient. Therefore, my overall rating is reject.

---

### Decision · Program_Chairs · 2023-01-20

**Decision:**

Reject

**Justification For Why Not Higher Score:**

The main issue was the lack of novelty, as it mostly applies existing time series prediction techniques without significant modification, but the writing issues and issues with the experiments as mentioned above were also of importance.

**Justification For Why Not Lower Score:**

N/A

**Metareview: Summary, Strengths And Weaknesses:**

The paper compares a number of deep learning-based and classical statistical approaches for predicting the number of tourists visiting a number of locations on an hourly basis. This is extended with an exploration of incorporating additional data (holidays, weather, tourist geolocation).

In general, reviewers agreed that studying tourism prediction using deep learning approaches is an interesting problem, but there are a few main issues, particularly:

- Novelty: the methodology has insufficient novelty, as it mostly applies existing time series prediction techniques without significant modification or novel insights.

- Experimental issues: the reviewer point out a number of issues for improvement, such as the statistical and practical significance of the improvements, some missing details (e.g. how the validation set was constructed), and the lack of ablation studies, as well as experiments going beyond 1-step ahead prediction.

- Writing issues: including typos, duplications, and need for clearer definitions.

The authors did not post a rebuttal during the review period. In the end, PCs and AC agreed that due to the above issues, the work is not ready for publication at ICLR, but the reviews offer a number of helpful suggestions for improvement, so I encourage the authors to continue improving the paper based on the reviews. The authors are also suggested to consider how to deal with privacy issues related to the use of individual tourist geolocation data.